# Effects of a Dog-Assisted Social- and Emotional-Competence Training for Prisoners: A Controlled Study

**DOI:** 10.3390/ijerph191710553

**Published:** 2022-08-24

**Authors:** Karin Hediger, Rahel Marti, Vivien Urfer, Armin Schenk, Verena Gutwein, Christine Dörr

**Affiliations:** 1Division of Clinical Psychology and Animal-Assisted Interventions, Faculty of Psychology, University of Basel, Missionsstrasse 62, 4056 Basel, Switzerland; 2Human and Animal Health Unit, Department of Epidemiology and Public Health, Swiss Tropical and Public Health Institute Basel, Kreuzstrasse 2, 4123 Allschwil, Switzerland; 3Faculty of Psychology, Open University, Postbus 2960, 6401 AT Heerlen, The Netherlands; 4Institute for Interdisciplinary Research on Human–Animal Interaction (IEMT), Kreuzstrasse 2, 4123 Allschwil, Switzerland; 5Pet-Agogik, Lärchenweg 12, 68804 Altlussheim, Germany; 6Department of Corrections Bruchsal, Schönbornstrasse 32, 76646 Bruchsal, Germany

**Keywords:** animal-assisted intervention, prison, social competence, emotional competence, dog

## Abstract

Background: Previous research has called for improving psychological interventions and developing new treatments for prisoners. Animal-assisted prison-based programmes have increasingly been used as an approach, but there is a lack of studies investigating the effectiveness of such programmes. Objective: To investigate the effects of a dog-assisted social- and emotional-competence training on the socioemotional competences of prisoners compared to treatment as usual. Methods: In a controlled trial, we investigated 62 prisoners that participated in either a 6-month dog-assisted psychotherapeutic programme or the standard treatment. We assessed social and emotional competences before and after the training and at a 4-month follow-up. Data were analysed with linear models. Results: The prisoners’ self-assessed social and emotional competences did not differ. The psychotherapists rated the prisoners’ emotional competences in the intervention group higher at the follow-up but not after the training. The psychotherapists did not rate the prisoners’ social competences in the intervention group differently but did find them to have higher self-regulation at follow-up and lower aggressiveness after the training than the control group. Conclusions: This study indicates that dog-assisted programmes with a therapeutic aim might be beneficial for prisoners. However, the inconsistent results indicate that more research is needed to determine the potential and limits of animal-assisted programmes in forensic settings.

## 1. Introduction

Currently, over 11 million people are held in prisons, and 30 million enter and leave custody each year [1]. There are various psychological treatments that aim to reduce criminal recidivism and to improve the social and emotional functioning of prisoners. The success of individual treatments in prisons largely depends on how treatments are designed and on the extent to which prisoners engage in the treatment process [2,3]. The relationship between the therapist and the person being treated is especially crucial for treatment success [4,5]. However, establishing a therapeutic alliance can be challenging due to prisoners’ problems with contact and communication behaviour, self-reflection, impulse control, and empathy [6]. Up to 50% of prisoners suffer from a mental illness such as depression, anxiety, posttraumatic stress disorder, substance-use disorder, and personality disorders [7,8,9]. Unfavourable bonding experiences, prolonged criminal careers, and distrust of the prison system as an institution and of the staff working there are other factors that make successful treatment difficult. As a result, prisoners’ motivation for treatment and change can be low [2]. Adapting treatments for prisoners such that they engage in them is thus often challenging.

The common structured therapeutic programmes developed for prisoners mainly use verbal and cognitive methods, for which there is often a lack of evidence in prison settings [10] and which have little effect on recidivism [11]. These programmes do not focus on direct experiences, and their application often requires that prisoners are already motivated to pursue change and treatment. Prisoners with poor social skills and comorbid mental disorders can only be reached to a limited extent by these conventional methods. For these reasons, there is strong call for improving psychological interventions and developing new treatment approaches for prisoners [11].

Animal-assisted prison-based programmes are one such alternative approach. Since the 1970s, various prisons in Europe, the USA, Canada, and Australia have increasingly been implementing animal-assisted programmes, mostly with dogs but also with horses and farm animals [12,13,14,15]. Studies have shown that dog-assisted programmes can improve prisoners’ social skills; increase trust, motivation, self-confidence, empathy, and emotion regulation; and reduce anxiety, depression, loneliness, impulsivity, and misconduct [16,17,18,19,20,21,22], but the results are mixed [17,23]. A large retrospective study on the effects of prison-based dog-training programmes found a decreased likelihood of re-arrest within 1 year [24], while a randomised controlled trial found no effects on empathy and behaviour problems in juveniles [25]. A meta-analysis on dog-training programmes that largely consisted of studies without a control group found a large effect of dog-training programmes on externalising outcomes such as recidivism, aggression, self-control, and institutional infractions and a small effect on internalising outcomes [21].

Most of the studies to date have investigated dog-training programmes in which prisoners train shelter dogs and prepare them for adoption or train them to become assistance dogs for people with disabilities or mental-health problems [15,26]. Such dog-training programmes are the most common form of prison-based dog programmes [12,13]. They have no explicit therapeutic aim and do not incorporate therapeutic techniques; instead, they are focused on training the dogs and the dogs’ well-being. Nevertheless, they are thought to contribute to rehabilitation and to improving psychological functioning, and they even seem to have a positive cost-benefit analysis [24,27,28].

Programmes that incorporate dogs in therapeutic interventions with the specific aim of facilitating therapeutic or educational goals, e.g., [19,29], are less well investigated. A recent meta-analysis included 11 studies on all types of prison-based dog programmes and found a small overall effect on recidivism and socioemotional functioning and a small-to-medium effect on recidivism only [12]. However, there remains a clear lack of studies investigating the effectiveness of including a dog in therapeutic programmes for prisoners.

The aim of this study is to investigate the effect of dog-assisted social- and emotional-competence group training for prisoners compared to treatment as usual. We hypothesised that the social and emotional competences of the prisoners would increase more from the beginning to the end of the training in the intervention group than in the control group. We expected that the prisoners’ self-worth would increase more while aggressivity, insecurity, and subjective stress would decrease more from the beginning to the end of the training compared to the control group. Moreover, we hypothesised that the prisoners would feel calmer and more awake and would have a better mood after the individual intervention sessions compared to the start of the sessions.

## 2. Materials and Methods

### 2.1. Participants

Sixty-two male prisoners participated in this study. All participants, aged 24 to 67 years (*M* = 42, *SD* = 11), were incarcerated for sexual and violent offenses. Participants’ offenses differed in severity and type, resulting in varying levels of imprisonment, with some subsequently facing mandatory security detention, if not life imprisonment. Prisoners were eligible for participation if they lived in the social therapy ward at Bruchsal Prison, Germany, or in the Hohenasperg Sociotherapeutic Institution, Germany, and provided informed written consent. Exclusion criteria were existing allergies to animal hair, significant fear of dogs, violent behaviour towards animals, or a stay at the facility for less than 8 months. A total of 30 participants were from Bruchsaal and 32 from Hohenasperg. The data were collected from 7 August 2017 to 18 May 2020. Figure 1 shows the CONSORT flowchart.

### 2.2. Standard Protocol Approvals, Registrations, and Participant Consents

The screening process included all prisoners in the two facilities at certain points in time. All participants provided written informed consent. The protocols were approved by the Criminological Service of the Department of Corrections for Baden-Württemberg, Germany. The animal-assisted therapy was performed according to the guidelines of the International Association of Human–Animal Interaction Organizations [30] to ensure patient safety and animal welfare. No sessions had to be ended early, and no adverse incidents occurred. After participating in the study (after the follow-up measurement), all participants had the possibility to continue with animal-assisted activities if they wished. Of the 27 prisoners who completed the programme, 17 continued with animal-assisted activities, 2 additional participants were interested in continuing, but organisational reasons made impossible for them to do so, 5 did not wish to continue, and for the remaining 3 we have no information.

### 2.3. Study Design and Procedures

The study had a quasi-experimental design with an intervention group and a control group receiving treatment as usual. In the original study protocol, we planned to randomly allocate participants to the conditions. However, randomisation was not possible due to organisational difficulties at the facilities.

All prisoners that lived in the selected wards were made aware of the study and the offer by their responsible psychologist. If they were interested in participating, prisoners could attend an information event held by a psychologist and a study-team member to clarify any questions or concerns. If they agreed to participate, prisoners gave their written informed consent. Assignment to the intervention group was determined based on the self-selection of the prisoners and the recommendations of the therapists. Assignment to the control group occurred based on self-selection but also through a waiting list in case the intervention group was already full. All study participants continued their individual standard treatment plan in addition to participating in the study (treatment as usual). These plans consisted of regular psychotherapeutic sessions and additional group treatments if indicated. The intervention group additionally received the dog-assisted social- and emotional-competence group training.

Self-assessments with questionnaires took place at baseline before the intervention (t_1_; week 0), posttreatment after 6 months (t_2_; week 24), and at a 4-month follow-up (t_3_; week 40). The prisoners received a compensation of EUR 10 if they completed the questionnaires at each of the measurement points. The intervention group additionally filled in a questionnaire before and after each intervention session as a process measure. The primary outcomes consisted also of the questionnaires, which were filled out in external assessments at baseline before the intervention (t_1_; week 0), posttreatment after 6 months (t_2_; week 24), and at a 4-month follow-up (t_3_; week 40). This was done by the responsible psychotherapists. The psychotherapists assessing the prisoners were not involved in the study but could not be blinded.

#### 2.3.1. Dog-Assisted Social- and Emotional-Competence Group Training

The intervention was a dog-assisted psychotherapeutic programme developed with the aim of increasing the socioemotional skills of prisoners in long-term custody. The programme was conducted by a psychotherapist working at the respective facility together with a specialist in animal-assisted interventions with a diploma in special education and the specialist’s dogs. The programme was manualised, lasted 6 months, and was divided into two modules: a basic module and a small-group module. A third practice module originally included in the manualised programme was not part of the intervention and evaluation of this study. The basic module consisted of eight weekly sessions of 2 h each in which groups of eight participants focused on the topics of nonverbal communication, body language, the roles of sender and receiver, criticism and feedback, and perception and interpretation. The goal was to develop an understanding of coherent communication in a total of 8 weeks. In the beginning, the dog mostly served to promote participants’ motivation for change and as a social catalyst. Exemplary exercises included: prisoners sat on chairs in a circle while the dog moved freely from one to another, greeting each prisoner; prisoners conveyed their mood by picking one of various pictures of the dog; prisoners led the dog nonverbally around three pylons and gave each other feedback at the end of the exercise.

After the participants became acquainted with the dog, the dog took on a more central role in the small-group module, where it was integrated more specifically in the exercises. The small-group module consisted of nine sessions for 2 h every second week over a period of 4 months for each group. Of these, seven sessions took place in the small groups, whereas the last two sessions were again with both groups together. For the small groups, participants in the basic module were split into two groups, leading to groups of four participants maximum. The groups were divided according to a clinical assessment for matching the participants. In the small-group module, the goal was to develop action strategies for the participants. The module was intended to involve the participants as actively as possible. The exercises were individualised as much as possible for each participant. For example, prisoners regulated the closeness and distance to the dog by nonverbally directing him to circulate along a line around them; the dog followed the prisoners through the room; prisoners directed the dog through a course consisting of different difficult obstacles; prisoners gave feedback to each other at the end of the exercises. This module further aimed to help the prisoners recognise their strengths and weaknesses as well as to develop goals for change. To increase empathy, the therapist asked the participants to evaluate the dog’s behaviour for motives and to observe the interaction of the other group members. The last two sessions in this module were held again in the whole group.

At the end of each session throughout the whole programme, each participant received a sheet for recording their experiences, what they learned, and what they worked on to reflect on and deepen.

#### 2.3.2. Dogs

Five dogs participated in the study. Four dogs were Belgian shepherds (female: *n* = 3; male, *n* = 1) aged between 9 years and 2 months and 1 year and 9 months at study start. One dog was a Pastor Garafiano crossbreed aged 5 years and 7 months. All the dogs had been trained and were experienced in animal-assisted interventions, had worked before with prisoners, and were trained for this specific setting.

The specialist in animal-assisted interventions was the dogs’ owner. She was always present during the programme and was responsible for the dogs’ well-being at all times. During the programme, the dogs had the possibility to retreat at any time to take a break for rest and relaxation. Two or three dogs worked in parallel with one group. There was an extra office room that was familiar to the dogs where they could stay if they needed to retreat. All the dogs had other fields of activity and an occupational or leisure programme to compensate for their work. The criterion for terminating a session was defined as too much impulsivity and open violence from the prisoners towards the dogs. No session had to be ended early.

### 2.4. Measures

For the primary and secondary outcomes, except for mood, the prisoners completed self-report questionnaires and the psychotherapists completed questionnaires at baseline (t_1_; week 0), posttreatment (t_2_; week 24), and follow-up-treatment (t_3_; week 40). Moreover, we had a process measurement where the prisoners completed a self-report questionnaire regarding their mood before and after each session.

#### 2.4.1. Primary Outcome Measure

**Emotional Competences**. We used the Emotional-Competence Questionnaire [31] to assess the prisoners’ emotional competences. It includes both a self-assessment and an external assessment. The questionnaire encompasses 62 items, which are rated on a 5-point Likert scale ranging from *not at all (1)* to *totally (5)*. The results include a total score for emotional competence and four subscales: recognition of own emotions; recognition of emotions in others; emotion regulation/control; emotional expressivity. The scale values range from 1 to 5, with higher values indicating higher emotional competence. With an additional 29 items, the questionnaire also assesses two additional subscales: regulation of others’ emotions; attitudes towards emotions. The internal consistency of the questionnaire ranges between 0.89 and 0.93.

**Social Competences**. We used the inventory for social competences 360° [32] to measure social competences. It includes both a self-assessment and an external assessment. The questionnaire consists of 32 items that are rated on a 5-point Likert scale ranging from *not at all (1)* to *totally (5)*. It has four subscales: social orientation, offensiveness, self-regulation, and reflexibility. Each subscale ranges between 1 and 5, with higher values indicating higher social competence. The questionnaires’ internal consistency ranges between 0.70 and 0.79 for the self-assessment and between 0.77 and 0.87 for the external assessment.

#### 2.4.2. Secondary Outcome Measures

**Empathy.** For the assessment of empathy, the Saarbrücken Personality Questionnaire, the German translation of Davis’s [33] Interpersonal Reactivity Index, was used. The questionnaire is a self-assessment and consists of 28 items that are rated on a 5-point Likert scale ranging from *never (1)* to *always (5)*. It has four subscales: perspective taking, fantasy, empathic concern, and personal distress, each ranging from 4 to 20. Fantasy, empathic concern, and personal distress are related to the emotional aspect of empathy, while the perspective-taking scale represents the cognitive-empathy factor [34,35]. The questionnaire’s internal consistency ranges between 0.66 and 0.78.

**Self-Esteem**. We used the revised German version of the Rosenberg Self-Esteem scale [36]. The questionnaire is a self-assessment and consists of 10 items that are rated on a 4-point Likert scale ranging from *not at all (0)* to *totally (3).* The total score (ranging between 0 and 30) reflects general self-esteem with higher values indicating higher self-esteem. The questionnaire has good psychometric properties with a medium internal consistency between 0.83 and 0.88.

**Aggressiveness**. We used the short-form Questionnaire for Aggressivity Factors [37] to assess the prisoners’ readiness to be aggressive. It consists of 49 items that are rated on a 6-point Likert scale ranging from *not at all (0)* to *totally (5)* and has the subscales: spontaneous aggressivity, reactive aggressivity, excitability, self-aggressivity, and aggression inhibition. A total score for externally oriented aggressivity can be calculated with values between 0 and 165 with higher values indicating higher aggressivity. Internal consistency of the total score is 0.89, while the internal consistencies of the subscales range between 0.55 and 0.84.

**Psychosocial Problems**. We used the Problem Questionnaire, a revised form of the Scale for Student Problems [38], to measure general psychic and somatic problems. The questionnaire comprises 40 items that are each rated on a 5-point Likert scale ranging from *not at all (1)* to *very strong (5).* The questionnaire has a total score and four subscales: social anxiety, vulnerability, depression, exhaustion, and a total score for psychosocial problems ranging from 40 to 200. Higher values indicate greater psychosocial problems. The internal consistency in the current sample of the total score is 0.94, while the internal consistencies of the subscales range between 0.79 and 0.84.

**Insecurity**. We used the Insecurity Questionnaire by Ullrich and Ullrich de Muynck [39] to measure social anxiety and social incompetence. The questionnaire is a self-assessment instrument with 65 items that are rated on a 6-point Likert scale ranging from *not at all (0)* to *exactly (5).* It consists of six subscales: anxiety of failure and criticism, contact anxiety, ability to demand, ability to say no, guilt, and decency, with scores ranging from 0 to 75 for each subscale. Internal consistency of the subscales ranges between 0.91 and 0.95.

**Mood**. The two parallel short forms of the German version of the Multidimensional Mood State Questionnaire (MDBF) [40] were used to measure mood before and after each intervention session. The questionnaire comprises 12 items that are each rated on a 5-point intensity scale. The questionnaire reveals results of three bipolar dimensions: good–bad mood, awake–tired, and calm–activated, ranging between the values 4 and 20. Higher values indicate participants are in a good mood, feel more awake, and feel calmer. The questionnaire is widely used and has good psychometric properties. The internal validity of the short form is between α = 0.73 and α = 0.89.

#### 2.4.3. Other Measures

**Demographics**. Descriptive data were collected on age, length of stay, centre of stay, and ICD-10 diagnoses. The diagnoses were derived from the psychiatric reports. If these were older than 2 years, the Structured Clinical Interview for DSM-IV was conducted.

### 2.5. Statistical Analysis

We estimated the mean and standard deviation of the primary outcome on the basis of a pilot study and expected a small effect size for the before–after comparison (*d* = 0.25). On this basis, the a priori sample size calculation with G*Power revealed a total sample size of 44 to detect an effect of 0.25 with a power of 80% at a significance level of 95%. We increased the final sample size to 62 to account for possible dropouts.

The primary outcomes (postintervention social and emotional competence) were analysed using linear models (analysis of covariance, ANCOVA) with the corresponding preintervention outcome as a covariate and the treatment condition as between-subject factor. The secondary outcomes (social and emotional competence at follow-up, all other measures except the MDBF at postintervention and at follow-up) were analysed in the same way. For these models, we used Levene’s test to determine the variance of homogeneity of the two conditions. The homogeneity of the regression slopes and the distribution of the variables were tested using Shapiro–Wilk’s test and Q–Q plots. The variables were visually checked for normality (histogram and Q–Q plot). Model diagnostics included visual checks for normality and the homogeneity of residuals. All data were approximately normally distributed, and requirements for the analyses were met.

To analyse the process-measure mood (MDBF), we used linear mixed models with the time and session number as fixed effects and the subjects as random effects. The variables were visually checked for normality (histogram and Q–Q plot). Model diagnostics included visual checks for normality and the homogeneity of residuals. All data were approximately normally distributed. Analyses were based on the intention-to-treat approach, which includes all available cases.

Analyses were based on available cases according to the principles of the intention-to-treat approach. Results were analysed and reported according to the CONSORT 2010 statement [41]. Data are presented as means and standard deviations. For all analyses, the mean difference (*difference*) was used as an effect size, the confidence interval was defined at 95%, and the significance level was set at 0.05. All statistical analyses were performed with R for Windows, version 4.1.0.

## 3. Results

### 3.1. Sample Characteristics

All recruited prisoners were between 24 and 67 years old (*M* = 42, *SD* = 11) and had already been at the prison at study start between 1 and 368 months (*M* = 38, *SD* = 55). One prisoner who was assigned to the intervention group had not filled in the questionnaires before the intervention and dropped out during the intervention, so he could not be included in the analyses (Figure 1). Nine prisoners dropped out after starting the intervention (*n* = 5) or after being assigned to the treatment-as-usual group (*n* = 4). Six of these dropouts were due to transfers to another department or institution, whereas two participants from the intervention group and one from the control group terminated participation due to personal reasons.

The final sample of 53 participants were between 24 and 67 years old (*M* = 41.66, *SD* = 11.41) and consisted of 27 prisoners in the intervention group (mean age in years (*SD*) = 38.07 (10.24)) and 26 prisoners in the control group (mean age in years (*SD*) = 45.38 (11.55)). At the start of the study, the participants had been at the prison for between 1 and 368 months (*M* = 39.91, *SD* = 58.86 months). A total of 25 prisoners were recruited in Bruchsaal and 27 in Hohenasperg. Most of the prisoners were diagnosed with a mixed or other personality disorder or a specific personality disorder (see Table 1) and with mental and behavioural disorders due to use of alcohol as well as with sexual-preference disorders.

There were no significant group differences in any of the primary outcomes at baseline (see Table 2, Table 3 and Table A1). In the secondary outcomes, significant differences between the groups at baseline were seen in excitability with higher values in the control group than in the intervention group (*p* = 0.024), in total psychosocial problems with higher values in the control group (*p* = 0.19), in social anxiety with higher values in the control group (*p* = 0.050), and in exhaustion with higher values in the control group (*p* = 0.008, see Table A1).

### 3.2. Primary Outcomes

**Emotional Competences, Self-Assessment.** Participants did not differ in their self-assessed overall emotional competences after the training compared to the control group (*difference* = 0.06, *CI* = −0.16 to 0.28, *p* = 0.597, see Table 2). Additionally, no difference was found at follow-up (*difference* = 0.04, *CI* = −0.21 to 0.29, *p* = 0.734). Prisoners in both groups did not differ in their ability to recognise their own emotions (postintervention: *difference* = 0.03, *CI* = −0.25 to 0.32, *p* = 0.806; follow-up: *difference* = 0.04, *CI* = −0.29 to 0.37, *p* = 0.805), in their ability to recognise emotions in others (postintervention: *difference* = −0.05, *CI* = −0.27 to 0.17, *p* = 0.676; follow-up: *difference* = −0.08, *CI* = −0.38 to 0.23, *p* = 0.623), in their ability to regulate emotions (postintervention: *difference* = 0.12, *CI* = −0.15 to 0.29, *p* = 0.365; follow-up: *difference* = 0.22, *CI* = −0.08 to 0.52, *p* = 0.140), or in emotional expressivity (postintervention: *difference* = 0.09, *CI* = −0.26 to 0.43, *p* = 0.607; follow-up: *difference* = −0.11, *CI* = −0.47 to 0.25, *p* = 0.550).

**Emotional Competences, External Assessment.** The psychotherapists did not rate the overall emotional competences of the prisoners in the intervention group after the training differently from those of the control group (*difference* = 0.19, *CI* = −0.17 to 0.56, *p* = 0.286, see Table 2). However, a significant difference was found at follow-up with higher ratings in the intervention group (*difference* = −0.34, *CI* = −0.67 to −0.01, *p* = 0.044). We found that the psychotherapists did not rate the prisoners in the intervention group differently compared to the control group after the treatment, but at follow-up they did tend to rate control-group participants’ ability to recognise their own emotions higher (postintervention: *difference* = −0.04, *CI* = −0.36 to 0.27, *p* = 0.786; follow-up: *difference* = −0.36, *CI* = −0.73 to 0.01, *p* = 0.056) and their ability to recognise emotions in others higher (postintervention: *difference* = −0.00, *CI* = −0.34 to 0.33, *p* = 0.982; follow-up: *difference* = −0.52, *CI* = −0.96 to −0.08, *p* = 0.022). The prisoners’ ability to regulate their emotions was rated significantly higher for the intervention group at follow-up than for the control group (postintervention: *difference* = 0.04, *CI* = −0.30 to 0.37, *p* = 0.828; follow-up: *difference* = −0.51, *CI* = −0.92 to −0.09, *p* = 0.018). No difference was found in the psychotherapists’ assessments of the prisoners’ emotional expressivity (postintervention: *difference* = 0.24, *CI* = −0.18 to 0.65, *p* = 0.257; follow-up: *difference* = −0.09, *CI* = −0.57 to 0.39, *p* = 0.700).

**Social Competences, Self-Assessment.** Participants did not differ in their self-assessed overall social competences after the training compared to the control group (*difference* = 0.00, *CI* = −0.14 to 0.15, *p* = 0.985, see Table 3). There was also no difference at follow-up (*difference* = −0.01, *CI* = −0.18 to 0.17, *p* = 0.932). The prisoners in both groups did not differ in their social orientation (postintervention: *difference* = −0.06, *CI* = −0.23 to 0.17, *p* = 0.51; follow-up: *difference* = −0.08, *CI* = −0.30 to 0.13, *p* = 0.440), offensiveness (postintervention: *difference* = 0.03, *CI* = −0.16 to 0.22, *p* = 0.751; follow-up: *difference* = −0.05, *CI* = −0.28 to 0.18, *p* = 0.666), self-regulation (postintervention: *difference* = 0.07, *CI* = −0.20 to 0.33, *p* = 0.607; follow-up: *difference* = −0.05, *CI* = −0.31 to 0.22, *p* = 0.716), or reflexivity (postintervention: *difference* = −0.11, *CI* = −0.40 to 0.18, *p* = 0.444; follow-up: *difference* = 0.06, *CI* = −0.23 to 0.34, *p* = 0.702).

**Social Competences, External Assessment.** The psychotherapists did not rate the overall social competences of the prisoners differently after the training compared to the control group (*difference* = −0.07, *CI* = −0.22 to 0.08, *p* = 0.336, see Table 3). There was also no difference at follow-up (*difference* = −0.16, *CI* = −0.41 to 0.10, *p* = 0.217). The psychotherapists tended to rate the social orientation of the prisoners in the intervention group higher compared to the control group (postintervention: *difference* = −0.22, *CI* = −0.47 to 0.03, *p* = 0.085; follow-up: *difference* = −0.31, *CI* = −0.67 to 0.05, *p* = 0.092), but the prisoners’ offensiveness did not differ (postintervention: *difference* = 0.07, *CI* = −0.17 to 0.31, *p* = 0.562; follow-up: *difference* = −0.01, *CI* = −0.37 to 0.34, *p* = 0.946). Self-regulation differed significantly at follow-up (postintervention: *difference* = −0.21, *CI* = −0.52 to 0.09, *p* = 0.164; follow-up: *difference* = −0.38, *CI* = −0.76 to −0.01, *p* = 0.043), but no effect was found in the psychotherapists’ ratings of the prisoners’ reflexibility (postintervention: *difference* = −0.06, *CI* = −0.34 to 0.23, *p* = 0.695; follow-up: *difference* = −0.16, *CI* = −0.49 to 0.28, *p* = 0.473).

### 3.3. Secondary Outcomes

**Empathy.** Prisoners in the intervention group did not differ in their self-assessed ability for perspective taking (postintervention: *difference* = −0.16, *CI* = −0.59 to 0.28, *p* = 0.473; follow-up: *difference* = −0.16, *CI* = −0.59 to 0.28, *p* = 0.473), fantasy (postintervention: *difference* = −0.60, *CI* = −2.33 to 1.13, *p* = 0.488; follow-up: *difference* = 0.90, *CI* = 1.06 to 2.85, *p* = 0.362), empathic concern (postintervention: *difference* = 0.31, *CI* = −0.80 to 1.42, *p* = 0.581; follow-up: *difference* = 0.82, *CI* = −0.95 to 1.59, *p* = 0.356), and personal distress (postintervention: *difference* = 0.78, *CI* = −1.04 to 2.60, *p* = 0.392; follow-up: *difference* = 0.23, *CI* = −1.69 to 2.15, *p* = 0.810) compared to the control group. The means and standard deviations at each time point are presented in Table A1 in Appendix A.

**Self-Esteem.** Prisoners in the intervention group did not differ in their self-assessed self-esteem compared to the control group after the intervention (*difference* = −0.25, *CI* = −2.53 to 2.04, *p* = 0.829) or at follow-up (*difference* = −0.20, *CI* = −2.43 to 2.03, *p* = 0.860). The means and standard deviations at each time point are presented in Table A1 in Appendix A.

**Aggressiveness.** Prisoners in the intervention group rated their total externally oriented aggressivity as significantly lower compared to the control group at postintervention (*difference* = 7.46, *CI* = 0.46 to 14.47, *p* = 0.037) but not at follow-up (*difference* = 6.78, *CI* = −1.79 to 15.35, *p* = 0.118). There was also a tendency for lower scores in their spontaneous aggressivity after the intervention but not at follow-up (postintervention: *difference* = 2.92, *CI* = −0.16 to 6.00, *p* = 0.062; follow-up: *difference* = 2.55, *CI* = −1.25 to 6.35, *p* = 0.182). Reactive aggressivity differed significantly (postintervention: *difference* = 3.38, *CI* = 0.25 to 6.51, *p* = 0.035; follow-up: *difference* = 4.80, *CI* = 0.10 to 8.61, *p* = 0.015), but excitability (postintervention: *difference* = 1.49, *CI* = −1.79 to 4.76, *p* = 0.366; follow-up: *difference* = −0.13, *CI* = −3.46 to 3.21, *p* = 0.939), self-aggression (postintervention: *difference* = 0.13, *CI* = −2.87 to 3.13, *p* = 0.931; follow-up: *difference* = 1.67, *CI* = −1.66 to 4.99, *p* = 0.318), and aggression inhibition (postintervention: *difference* = −0.98, *CI* = −4.27 to 2.31, *p* = 0.551; follow-up: *difference* = −0.36, *CI* = −3.26 to 2.54, *p* = 0.803) did not. The means and standard deviations at each time point are presented in Table A1 in Appendix A.

**Psychosocial Problems.** Prisoners in the intervention group did not differ in the number of psychosocial problems compared to the control group after the intervention (*difference* = −0.97, *CI* = −12.58 to 10.64, *p* = 0.865) or at follow-up (*difference* = −0.35, *CI* = −9.86 to 9.16, *p* = 0.980). No differences were found in the subscales social anxiety (postintervention: *difference* = 1.54, *CI* = −1.34 to 4.49, *p* = 0.296; follow-up: *difference* = −0.27, *CI* = −2.97 to 2.24, *p* = 0.840), vulnerability (postintervention: *difference* = 0.29, *CI* = −2.29 to 2.88, *p* = 0.820; follow-up: *difference* = 1.27, *CI* = −1.06 to 3.61, *p* = 0.278), depression (postintervention: *difference* = −0.25, *CI* = −3.74 to 3.24, *p* = 0.887; follow-up: *difference* = 1.11, *CI* = −2.15 to 4.37, *p* = 0.497), and exhaustion (postintervention: *difference* = 2.45, *CI* = −0.70 to 5.60, *p* = 0.125; follow-up: *difference* = 2.54, *CI* = −0.67 to 5.74, *p* = 0.118). The means and standard deviations at each time point are presented in Table A1 in Appendix A.

**Insecurity.** The prisoners in the intervention group did not differ from the control group in their anxiety of failure and criticism (postintervention: *difference* = 2.97, *CI* = −1.29 to 7.24, *p* = 0.167; follow-up: *difference* = 3.67, *CI* = −2.09 to 9.44, *p* = 0.206), contact anxiety (postintervention: *difference* = −0.23, *CI* = −5.19 to 4.73, *p* = 0.926; follow-up: *difference* = 0.60, *CI* = −5.12 to 6.32, *p* = 0.833), ability to demand (postintervention: *difference* = 2.65, *CI* = −1.54 to 6.83, *p* = 0.209; follow-up: *difference* = −0.68, *CI* = −4.97 to 3.60, *p* = 0.750), ability to say no (postintervention: *difference* = 0.18, *CI* = −5.61 to 5.98, *p* = 0.950; follow-up: *difference* = 2.11, *CI* = −3.81 to 8.03, *p* = 0.477), guilt (postintervention: *difference* = −1.03, *CI* = −8.14 to 6.07, *p* = 0.771; follow-up: *difference* = = 3.78, *CI* = −2.07 to 9.63, *p* = 0.200), or decency (postintervention: *difference* = −1.42, *CI* = −7.63 to 4.78, *p* = 0.647; follow-up: *difference* = 2.85, *CI* = −3.00 to 8.69, *p* = 0.332). The means and standard deviations at each time point are presented in Table A1 in Appendix A.

**Mood.** The prisoners in the intervention group indicated they were in a worse mood at the end of the training sessions compared to before the sessions (*difference* = −1.96, *CI* = −2.26 to −1.67, *p* < 0.01), and the session number did not influence this (*difference* = −0.01, *CI* = −0.04 to 0.02, *p* = 0.432). The prisoners felt less awake after the sessions (*difference* = −2.10, *CI* = −2.42 to −1.79, *p* < 0.001) with no change over time (*difference* = 0.01, *CI* = −0.02 to 0.04, *p* = 0.521) and less calm (*difference* = −4.67, *CI* = −5.00 to −4.33, *p* < 0.001) with no change over time (*difference* = 0.01, *CI* = −0.03 to 0.05, *p* = 0.576). The means and standard deviations at each time point are presented in Table A2 in Appendix A.

## 4. Discussion

The prisoners who participated in the dog-assisted social- and emotional-competence group training did not assess their social and emotional competences as higher after the training compared to prisoners in the control group who received treatment as usual. However, the psychotherapists rated the emotional competences of the prisoners in the intervention group higher at follow-up. Moreover, the psychotherapists rated prisoners in the intervention group to have higher self-regulation at follow-up and lower aggressiveness right after the training but not at follow-up. We did not find any effects of the programme on the prisoners’ empathy, self-esteem, psychosocial problems, or insecurity. Prisoners in the intervention group indicated that they were in a worse mood, felt less awake, and felt more activated after the individual training sessions compared to before them.

The fact that we found a beneficial effect of the dog-assisted training for the primary outcome of emotional competences and one aspect of social competence in the assessments of the psychotherapists but not in the prisoner’s self-assessments somewhat contradicts a recent meta-analysis of 11 controlled studies with 3013 participants [12]. The meta-analysis found a small effect on social-emotional functioning and suggested that prison-based dog programmes resulted in a small-to-medium effect in reducing criminal recidivism. Also, previous studies have linked prison-based dog programmes with enhanced communication, social skills, and empathy, more positive emotions, improved mood and higher self-esteem, and lowered depression and anxiety [16,19,21,42,43].

We found no effect of the training on the secondary outcomes of empathy, self-esteem, psychosocial problems, and insecurity. This is in line with other studies that found no or only negligible effects of animal-assisted programmes on prisoners’ empathy, self-efficacy, or self-esteem [17,23,25,42,44,45]. However, we found that the prisoners’ total aggressivity as well as reactive aggressivity were lower after the training in the intervention group compared to the control group, but this difference disappeared at follow-up.

Moreover, we found effects on the prisoners’ mood, which indicates that the individual sessions were activating but also strenuous and led to a reduction in good mood. This is contrary to our hypothesis at the beginning of the study and contrary to previous findings that reported more positive emotions and improved mood as effects of prison-based dog programmes [19,21]. However, we measured this outcome as a process indicator before and after each session and not before and after the whole programme, so the results cannot be directly compared.

The positive effects that we found on total emotional competence, on recognising emotions in others, on regulating emotions, and on the social competence of self-regulation were not seen right after the training but only at follow-up. This was 10 months after the start of the programme and 4 months after the end of the programme, which suggests that changes in prisoners’ social and emotional functioning might require some time. Moreover, we only found effects in the assessments by the psychotherapists and not in the prisoners’ self-assessments. It is unclear if this was due to possible difficulties the prisoners had in assessing their own conditions and skills, which would possibly make their self-assessments less accurate evaluations. It might also be possible that the prisoners who participated in the training were more critical in assessing their own skills because it was an important part of the programme to train self-evaluation, which could have led to a more realistic perspective of oneself. This might also be an explanation for the result that the prisoners rated their mood as lower after the individual sessions compared to before the sessions. Another possibility might be that the psychotherapists who did the assessments were biased by the prisoners’ participation in the programme. However, they were not involved in the study. In any case, this discrepancy in assessment types is an interesting finding that needs to be further addressed in future research.

Interestingly, qualitative research indicates a multitude of positive effects of prison-based dog-assisted programmes such as increased security, trust, mood, empathy, self-control, responsibility, sense of purpose and meaning, and more positive interactions in prisoners [20,28,46,47,48,49,50], which somewhat contrasts with the findings of the recent meta-analysis mentioned above [12] and the results of our study. For example, recent qualitative interviews with the staff of a prison-based dog-training programme suggested that the participants found purpose and meaning through participation in the programme, developed skills, enhanced their concept of self and perceived control, and showed greater community engagement [22]. For example, the programme changed the participants’ perspective such that they also wanted to change things in other areas of their lives [22]. These differences between quantitative and qualitative outcomes suggest that future research should carefully test what measures can target what effects and if they are suitable for prisoners. Moreover, what processes are relevant and where possible effects might be seen in prisoners need to be investigated. Research has shown that animal-assisted interventions in general can have a calming effect, promote social behaviour, facilitate learning processes, increase motivation and well-being, and be effective in treating patients with psychiatric disorders such as depression, schizophrenia, addiction, and posttraumatic-stress disorders [51,52,53,54,55]. This leads us to the hypothesis that the possible effects of animal-assisted programmes for prisoners might be reflected in other factors, such as the motivation to attend the sessions.

It has often been argued that the prison-based dog-training programmes in past studies might not have led to effects in psychosocial outcomes because they did not aim to be therapeutic but were rather intended to teach prisoners dog-training skills and were directed towards training dogs and the dogs’ well-being. In this study, the training directly targeted emotional and social skills and was conducted by a psychotherapist. We found beneficial effects in some of the trained skills but not in others. It thus remains unclear whether such programmes can indeed improve emotional and social competences and whether there are processes other than better socioemotional functioning that might be responsible for the positive effect of prison-based dog-training programmes on criminal recidivism. For example, it has been suggested that dog-assisted programmes might help prisoners become more receptive to treatments targeting criminal recidivism or help them build an alternative “anticriminal” identity [12]. Only a few studies have addressed the possible working mechanisms of these programmes [22]. Some authors have suggested that interacting with a dog might help prisoners engage in altruistic activities and thereby promote a prosocial identity and prosocial behaviours [46], or that it might help limit the strains of imprisonment [24]. Previous research has also shown that the presence of dogs in a prison can create a positive atmosphere in the facility, facilitate social interactions among prisoners as well as between prisoners and staff, and improve the institutional climate [48,49]. There thus might be spill-over effects from a group interacting with dogs to other prisoners who are not in the group. This is a hypothesis that must be tested carefully in the future, in particular, such processes should be taken into account in studies evaluating prison-based dog programmes.

### Limitations, Strengths, and Future Research

A limitation of the study was the lack of randomisation. Even though the baseline data did not differ significantly between the groups in most of the outcomes, other differences between the participants of the two groups may not have been adequately controlled. The intervention group had a slightly higher functioning level regarding excitability, psychosocial problems, social anxiety, and exhaustion compared to the control group, which might have influenced the results. Since self-selection could have led to a higher motivation for the training in the intervention group, it could be of interest to see that, in such a self-selected sample, we found some beneficial effects on emotional competences, self-regulation, and aggressiveness. However, the reasons why the prisoners chose to enrol in the programme are not clear, so it is not possible to draw a conclusion about this effect. Moreover, the prisoners could not be blinded, as it was obvious to them what group they were in, and the psychotherapists who filled in the questionnaires also could not be blinded. They were not, however, involved in the study, but the lack of blinding must be taken into account when interpreting the results. The control group only received treatment as usual with no additional matched increase in the amount of control treatment. The effects could thus be nonspecific to the type of intervention and might reflect rather a consequence of the overall increased amount of time spent in treatment. We also did not assess the mood of the control group before and after the therapy sessions. Thus, it is not possible to compare the effects between the groups and draw conclusions if they are specific to the dog-assisted social- and emotional-competence group training. In the intervention group, two people conducted the sessions (a psychotherapist and a specialist in animal-assisted interventions). The second person was not present during the regular psychotherapeutic session (treatment as usual) in the control condition. Moreover, the self-assessments of the social and emotional competences must be interpreted especially carefully, as they were not validated for prisoners and included relatively complex items that might have been challenging to answer for this target group. Additionally, the results of the subscales of the questionnaires measuring empathy and aggression must be interpreted with caution as the internal consistency of some subscales is below 0.70.

The inclusion of a control group, a follow-up assessment, and multi-informant outcomes as well as a multicentre approach were strengths of the study. Moreover, it is one of the first studies to assess the impact of a dog-assisted training for prisoners designed specifically with psychotherapeutic aims. We used a highly structured manual that was developed for this programme so that the intervention was comparable among the groups.

Future research should try to use questionnaires specifically designed for this population, to combine different sources of assessment, to employ behavioural observations or physiological correlates, and to include qualitative measures. We also suggest additionally focusing on process measures such as motivation rather than on symptom reduction. Further, prisoner characteristics that could predict the effects of dog-assisted training should be investigated since little is known about the responses of subgroups [56].

In different populations, it has been found that some humans profit more than others from including animals in therapeutic interventions [57]. Thus, previous pet ownership, attitudes towards dogs, and other sample characteristics should be taken into account. Moreover, the quality of the relationship between the dog and the prisoners might also influence the effects and should be quantified in future studies. In general, the working mechanisms of prison-based dog-assisted programmes should be addressed. To do so, other study designs such as N-of-1 trials might be useful. However, larger randomised controlled trials with blinded external assessors are also clearly needed. Future trials should also include an active control condition to match the total time spent in treatment for the intervention and the control groups. Since some studies have found that integrating animals into therapeutic programmes has add-on effects and other studies have shown animal-assisted programmes to be equivalently effective as evidence-based programmes [51], an interesting question for future research may be to investigate differential effects to see which people profit from animal assistance and which people do not need an animal as an addition to conventional treatments in order to benefit [56].

## 5. Conclusions

We found that the dog-assisted social- and emotional-competence group training did not have any effects on prisoners’ self-assessments of their social and emotional competences. The psychotherapists rated the prisoners’ emotional competences and self-regulation higher at follow-up, and the prisoners rated their aggressiveness to be lower after the training compared to the control group. The individual sessions led to a reduction in good mood and activated the prisoners in the intervention group. This study indicates that dog-assisted programmes with a specifically therapeutic aim might be beneficial for prisoners. However, more robust research on the effectiveness of animal-assisted programmes is needed to determine the potential and limits of this approach in correctional settings. Potential moderators should be investigated with a focus on identifying individuals who benefit from animal-assisted interventions more than from standard treatments.

## Figures and Tables

**Figure 1 ijerph-19-10553-f001:**
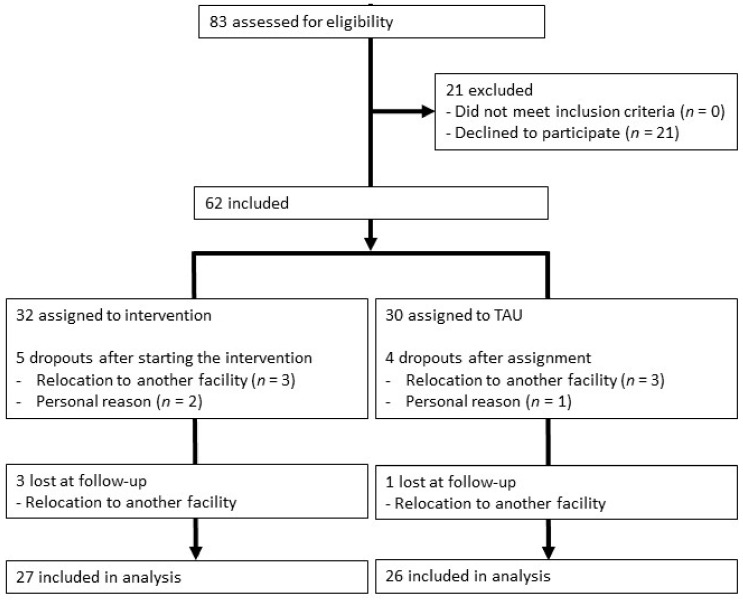
CONSORT flow chart.

**Table 1 ijerph-19-10553-t001:** Diagnoses of the study population (several diagnoses per prisoner possible).

Code	ICD-10 Category	Total	%
F61	Mixed and other personality disorders	29	19.59
F60	Specific personality disorders	23	15.54
F10	Mental and behavioural disorders due to use of alcohol	22	14.86
F65	Disorders of sexual preference	22	14.86
F91	Conduct disorders	9	6.08
F33	Recurrent depressive disorder	5	3.38
F63	Habit and impulse disorders	4	2.70
F19	Mental and behavioural disorders due to use of multiple drugs and use of other psychoactive substances	4	2.70
F14	Mental and behavioural disorders due to use of cocaine	4	2.70
F84	Pervasive developmental disorders	3	2.03
F13	Mental and behavioural disorders due to use of sedatives or hypnotics	3	2.03
F12	Mental and behavioural disorders due to use of cannabinoids	2	1.35
F52	Sexual dysfunction, not caused by organic disorder or disease	2	1.35
F19	Mental and behavioural disorders due to use of multiple drugs and use of other psychoactive substances	2	1.35
F90	Hyperkinetic disorders	2	1.35
F34	Persistent mood (affective) disorders	1	0.68
F31	Bipolar affective disorder	1	0.68
F11	Mental and behavioural disorders due to use of opioids	1	0.68
F79	Unspecified mental retardation	1	0.68
F68	Other disorders of adult personality and behaviour	1	0.68
F40	Phobic anxiety disorders	1	0.68
F43	Reaction to severe stress and adjustment disorders	1	0.68
F32	Depressive episode	1	0.68
F66	Psychological and behavioural disorders associated with sexual development and orientation	1	0.68
F70	Mild mental retardation	1	0.68
Z73	Burnout	1	0.68
Q98	Other sex-chromosome abnormalities, male phenotype, not classified elsewhere	1	0.68

**Table 2 ijerph-19-10553-t002:** Development of emotional competences over time for the intervention and the control group.

Outcome	Scale	Time Point	Intervention *n*	Intervention *M* (*SD*)	Control *n*	Control *M* (*SD*)
Self-assessment	Recognition of own emotions	t1	27	3.58 (0.70)	26	3.21 (0.78)
		t2	27	3.68 (0.71)	24	3.46 (9.41)
		t3	24	3.78 (0.76)	26	3.57 (0.65)
	Recognition of emotions in others	t1	27	3.50 (9.73)	26	3.28 (0.50)
		t2	27	3.52 (0.64)	24	3.30 (0.47)
		t3	24	3.61 (0.72)	26	3.33 (0.67)
	Emotion regulation/control	t1	27	3.52 (9.67)	26	3.19 (0.68)
		t2	27	3.54 (9.77)	24	3.33 (9.58)
		t3	24	3.53 (9.78)	26	3.53 (9.61)
	Emotional expressivity	t1	27	2.83 (9.75)	26	2.62 (9.62)
		t2	27	3.05 (0.88)	24	2.92 (9.69)
		t3	24	3.15 (0.76)	26	2.91 (9.68)
	Total score	t1	27	3.36 (9.60)	26	3.08 (0.48)
		t2	27	3.45 (9.62)	24	3.25 (9.39)
		t3	24	3.52 (0.60)	26	3.33 (0.54)
External assessment	Recognition of own emotions	t1	27	2.92 (0.72)	26	2.97 (0.87)
		t2	25	3.15 (0.71)	24	3.06 (0.95)
		t3	22	3.18 (0.82)	23	2.80 (0.83)
	Recognition of emotions in others	t1	27	2.70 (0.63)	26	2.55 (0.91)
		t2	25	2.96 (0.80)	24	2.67 (1.05)
		t3	22	3.20 (0.90)	23	2.51 (0.92)
	Emotion regulation/control	t1	27	2.83 (0.63)	26	2.51 (1.04)
		t2	25	2.88 (0.83)	24	2.63 (1.08)
		t3	22	3.20 (0.81)	23	2.42 (0.96)
	Emotional expressivity	t1	27	2.51 (0.74)	26	2.48 (0.59)
		t2	25	2.66 (0.86)	24	2.82 (0.92)
		t3	22	2.78 (0.97)	23	2.63 (9.99)
	Total score	t1	27	2.74 (0.49)	26	2.63 (9.73)
		t2	25	2.91 (0.66)	24	2.79 (0.88)
		t3	22	3.09 (0.74)	23	2.83 (0.80)

*n* = number, *M* = mean, *SD* = standard deviation.

**Table 3 ijerph-19-10553-t003:** Development of social competences over time for the intervention and the control group.

Outcome	Scale	Time Point	Intervention *n*	Intervention *M* (*SD*)	Control *n*	Control *M* (*SD*)
Self-assessment	Social orientation	t1	26	3.41 (0.46)	26	3.44 (0.46)
		t2	27	3.33 (0.54)	25	3.36 (0.58)
		t3	24	3.45 (0.55)	26	3.42 (0.62)
	Offensiveness	t1	26	3.12 (0.69)	26	3.13 (9.59)
		t2	27	3.10 (0.75)	25	3.14 (9.57)
		t3	24	3.24 (0.84)	26	3.16 (0.59)
	Self-regulation	t1	26	3.32 (9.76)	26	2.95 (9.56)
		t2	27	3.27 (0.80)	25	3.10 (0.55)
		t3	24	3.39 (0.73)	26	3.03 (0.63)
	Reflexibility	t1	26	3.52 (0.65)	26	3.54 (0.47)
		t2	27	3.43 (0.73)	25	3.39 (0.53)
		t3	24	3.47 (0.74)	26	3.52 (0.52)
	Total score	t1	26	3.34 (9.45)	26	3.27 (0.38)
		t2	27	3.28 (0.56)	25	3.25 (0.45)
		t3	24	3.39 (0.57)	26	3.28 (0.44)
External assessment	Social orientation	t1	27	2.77 (0.53)	26	2.47 (0.96)
		t2	25	3.00 (0.63)	24	2.51 (0.95)
		t3	22	3.19 (0.65)	23	2.63 (0.89)
	Offensiveness	t1	27	2.78 (9.78)	26	2.84 (0.59)
		t2	26	2.84 (0.60)	24	2.92 (0.70)
		t3	22	2.92 (0.85)	23	2.94 (0.81)
	Self-regulation	t1	27	2.78 (0.70)	26	2.39 (0.86)
		t2	25	2.30 (0.63)	24	2.51 (0..83)
		t3	22	3.22 (0.65)	23	2.56 (0.77)
	Reflexibility	t1	27	2.87 (0.68)	26	2.72 (9.73)
		t2	25	3.18 (0.88)	24	2.85 (0.77)
		t3	22	3.31 (0.96)	23	2.93 (0.91)
	Total score	t1	27	2.89 (0.39)	26	2.61 (0.60)
		t2	25	3.00 (0.48)	24	2.70 (0.58)
		t3	22	3.16 (0.58)	23	2.77 (0.58)

*n* = number, *M* = mean, *SD* = standard deviation.

## Data Availability

The data are available on the Harvard Dataverse database at https://doi.org/10.7910/DVN/O2K5DC.

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
