# Peer review of "Effects of a Dog-Assisted Social- and Emotional-Competence Training for Prisoners: A Controlled Study"

_ijerph, 2022, doi:10.3390/ijerph191710553_

Round 1

Reviewer 1 Report

The paper aims to investigate the effects of a dog-assisted social- and emotional-competence training on the socioemotional competences of prisoners. Overall, this aimed to fill a gap in controlled studies that evaluated a standardized and therapy directed dog-assisted treatment for prisoners. The careful selection of questionnaires targeted for specific evaluation of social and emotional competencies and the control study design are strengths of this work.

This study was designed with a quasi-experimental approach in a prospective, non-blinded, non-randomized controlled trial. Limitations of this design are discussed by the authors in section 4.1 however do not incorporate all the associated limitations and implications for interpretation of the results.

The design included a standard treatment in both the control and intervention groups [Lines 128-129] and an added dog-assisted treatment for the intervention group but no additional time-matched extension of standard treatment in the control group. Therefore, the intervention group received an overall increase in the collective duration of treatment as compared to the intervention group regardless of the type of intervention used which can be a significant factor in interpretation of the results. Any effect seen may be non-specific to the type of intervention and rather an effect of the overall longer duration of time spent in treatment. This is a significant limitation of the study not identified or explained by the authors and should be further discussed in the limitations section of the article if the control group was indeed not duration-matched.

As written, the MDBF to assess mood was only conducted on the intervention group which limits the interpretation of the results. Please provide clarification as to why this evaluation was only conducted in the intervention group and not included in the control group assessments. This is a limitation in interpretation of results in this particular evaluation and is not identified or explained by the authors and should be further discussed in the limitations section of the article.

Authors report prisoners were offered an opportunity to self-elect to continue with the dog-assisted therapy [Lines 114-115]. Please provide the results of this self-elected continuation of additional treatment and if this was conducted post-treatment during the four-month period before follow up (t3). 

Please provide an explanation and discussion on handling of interpretation on the questionnaire for evaluation of empathy and aggressiveness which were reported with lower internal consistency ratings (below 0.7). Additionally please provide a rating for the internal consistency on the Psychosocial problems questionnaire or provide explanation for not being reported. 

Minor comments:

[Line 441] There is an errant “the” in the sentence, “… group indicated the they were in a worse mood…”

[Line 466] Discussion indicates the results of this study somewhat contradicts the results of the 2020 meta-analysis study by Duindam et. al, but refers to the effects of dog-assisted programs on recidivism which is not a metric evaluated or reported in this study. Please re-phase to clarify which results of the 2020 metanalysis are contradicting the results of this study.

Author Response

Reviewer 1

The paper aims to investigate the effects of a dog-assisted social- and emotional-competence training on the socioemotional competences of prisoners. Overall, this aimed to fill a gap in controlled studies that evaluated a standardized and therapy directed dog-assisted treatment for prisoners. The careful selection of questionnaires targeted for specific evaluation of social and emotional competencies and the control study design are strengths of this work.

This study was designed with a quasi-experimental approach in a prospective, non-blinded, non-randomized controlled trial. Limitations of this design are discussed by the authors in section 4.1 however do not incorporate all the associated limitations and implications for interpretation of the results.

  1. The design included a standard treatment in both the control and intervention groups [Lines 128-129] and an added dog-assisted treatment for the intervention group but no additional time-matched extension of standard treatment in the control group. Therefore, the intervention group received an overall increase in the collective duration of treatment as compared to the intervention group regardless of the type of intervention used which can be a significant factor in interpretation of the results. Any effect seen may be non-specific to the type of intervention and rather an effect of the overall longer duration of time spent in treatment. This is a significant limitation of the study not identified or explained by the authors and should be further discussed in the limitations section of the article if the control group was indeed not duration-matched.

Thank you for this important comment. The control group was duration-matched in terms of pre/post/follow-up measurements. All measurements were made at the same time for each two matched groups. However, you are right that regarding the total of minutes of received therapy, the two groups differed. We included this in the limitation section and suggested that future studies make sure to control for this aspect. We have added:

Line 558: The control group only received treatment as usual with no additional matched increase in the amount of control treatment. The effects could thus be nonspecific to the type of intervention and might reflect rather a consequence of the overall increased amount of time spent in treatment.

Line 602: Future trials should also include an active control condition to match the total time spent in treatment for the intervention and the control groups.

  1. As written, the MDBF to assess mood was only conducted on the intervention group which limits the interpretation of the results. Please provide clarification as to why this evaluation was only conducted in the intervention group and not included in the control group assessments. This is a limitation in interpretation of results in this particular evaluation and is not identified or explained by the authors and should be further discussed in the limitations section of the article.

Thank you for this comment. Unfortunately, it was not possible to assess mood with the MDBF in the treatment as usual due to organisational reasons in the prison. We included this aspect in the limitations section with the following text:

Line 572: We also did not assess the mood of the control group before and after the therapy sessions. Thus, it is not possible to compare the effects between the groups and draw conclusions if they are specific to the dog-assisted social- and emotional-competence group training.

  1. Authors report prisoners were offered an opportunity to self-elect to continue with the dog-assisted therapy [Lines 114-115]. Please provide the results of this self-elected continuation of additional treatment and if this was conducted post-treatment during the four-month period before follow up (t3). 

Thanks for highlighting this interesting point. This possibility was only offered after the follow-up measurement at the end of the study. We extended this point in the text:

Line 115: After participating in the study (after the follow-up measurement), all participants had the possibility to continue with animal-assisted activities if they wished. Of the 27 prisoners who completed the program, 17 continued with animal-assisted activities, two additional participants were interested in continuing, but organisational reasons made impossible for them to do so, five did not wish to continue, and for the remaining three we have no information.

  1. Please provide an explanation and discussion on handling of interpretation on the questionnaire for evaluation of empathy and aggressiveness which were reported with lower internal consistency ratings (below 0.7). Additionally please provide a rating for the internal consistency on the Psychosocial problems questionnaire or provide explanation for not being reported. 

Thank you for pointing out this important aspect. For the psychosocial problems questionnaire, there is no internal consistency that can be found in the literature. Thus, we calculated the actual consistency in our current sample and reported it in the methods section as follows:

Line 257: The internal consistency in the current sample of the total score is 0.94, while the internal consistencies of the subscales range between 0.79 and 0.84.

Moreover, we discussed this aspect in the limitations section as follows:

Line 580: Also, the results of the subscales of the questionnaires measuring empathy and aggression must be interpreted with caution as the internal consistency of some subscales is below 0.70.

  1. [Line 441] There is an errant “the” in the sentence, “… group indicated thethey were in a worse mood…”

Thank you for this comment, you are right. We deleted the errant “the” in Line 450 (see track changes in manuscript).

  1. [Line 466] Discussion indicates the results of this study somewhat contradicts the results of the 2020 meta-analysis study by Duindam et. al, but refers to the effects of dog-assisted programs on recidivism which is not a metric evaluated or reported in this study. Please re-phase to clarify which results of the 2020 metanalysis are contradicting the results of this study.

The contradiction refers to the fact that in the 2020 meta-analysis by Duindam et al. no effect on social-emotional functioning was found whereas we found effects in social and emotional competences.

We rephrased it to make it clearer to what we refer as follows:

Line 470: The fact that we found a beneficial effect of the dog-assisted training for the primary outcome of emotional competences and one aspect of social competence in the assessments of the psychotherapists but not in the prisoner’s self-assessments somewhat contradicts a recent meta-analysis of 11 controlled studies with 3013 participants. The meta-analysis found no effect on social-emotional functioning, whereas it suggested that prison-based dog programmes resulted in a small-to-medium effect in reducing criminal recidivism.

Reviewer 2 Report

Great paper - well written and easy to follow, discussion places the findings into the context of the literature well and answered all of my 'but what about' questions.

I only have three comments:

1. Ethics - under the 'Institutional Review Board Statement' I presume the approval from the 'Criminological Service for Department of Corrections' serves as human ethics approval. What animal ethics approval process was implemented?

2. Language - just a minor (and personal) comment, in places the manuscript refers to the dogs being 'used' in prison contexts before. This frames the therapy dogs as nothing more than tools, please consider using different terminology.

3. The only 'confound' that wasn't mentioned in the Discussion (I may have missed it) was the extra person present (dog owner/handler) in the AAT sessions. For the TAU group they would have only had contact with their normal psychologist I presume? This is not a fatal flaw by any means - just one I would have like to see acknowledged.  

Author Response

  1. Ethics - under the 'Institutional Review Board Statement' I presume the approval from the 'Criminological Service for Department of Corrections' serves as humanethics approval. What animalethics approval process was implemented?

Thank you for this comment. Indeed, this served as human ethics board. We had no animal ethics approval process as this was not needed for this study but adhered to the International Association of Human–Animal Interaction Organizations (IAHAIO) standards.  

  1. Language - just a minor (and personal) comment, in places the manuscript refers to the dogs being 'used' in prison contexts before. This frames the therapy dogs as nothing more than tools, please consider using different terminology.

Thank you for this very important comment. We totally agree with you and changed the terminology. In line 187 it now reads: All the dogs had been trained and were experienced in animal-assisted interventions, had been working before with prisoners, and were trained for this specific setting.

  1. The only 'confound' that wasn't mentioned in the Discussion (I may have missed it) was the extra person present (dog owner/handler) in the AAT sessions. For the TAU group they would have only had contact with their normal psychologist I presume? This is not a fatal flaw by any means - just one I would have like to see acknowledged.  

Thank you for this comment, that’s an important point. We included this in the discussion section under limitations as follows:

Line 555: In the intervention group two people conducted the sessions (psychotherapist and a specialist in animal-assisted interventions). The second person was not present during the regular psychotherapeutic session (treatment as usual) in the control condition.

Round 2

Reviewer 1 Report

The new version of the manuscript provides the necessary additions and revisions to adequately addressed the limitations of the study design and conclusions.